# The Core and Distinction of the Gut Microbiota in Chinese Populations across Geography and Ethnicity

**DOI:** 10.3390/microorganisms8101579

**Published:** 2020-10-14

**Authors:** Deng Lin, Ran Wang, Junjie Luo, Fazheng Ren, Zhenglong Gu, Yiqiang Zhao, Liang Zhao

**Affiliations:** 1Beijing Advanced Innovation Center for Food Nutrition and Human Health, Department of Nutrition and Health, China Agricultural University, Beijing 100083, China; apiaolin@163.com (D.L.); wangran8609@126.com (R.W.); luojj@pku.edu.cn (J.L.); renfazheng@cau.edu.cn (F.R.); 2Key Laboratory of Precision Nutrition and Food Quality, Ministry of Education, College of Food Science and Nutritional Engineering, China Agricultural University, Beijing 100083, China; 3Division of Nutritional Sciences, Cornell University, Ithaca, NY 14853, USA; zg27@cornell.edu; 4Key Laboratory of Functional Dairy, College of Food Science and Nutritional Engineering, China Agricultural University, Beijing 100083, China

**Keywords:** gut microbiota, Chinese populations, geography, ethnicity, Han nationality

## Abstract

The diversity of the human gut microbiota constitutes a fundamental health indicator of different populations. The relative importance of geographical location and ethnicity on the gut microbiota, however, has not been previously addressed. Due to unique ethnic distributions across China, we recruited distinct minority ethnic groups, including Han populations, in each of the seven cities that were explored in this study. We investigated the gut microbiota of 394 healthy subjects (14 groups) from these seven different cities using 16S rRNA sequencing. Our results indicated that both geographical location and ethnicity were major factors. However, geographical location exhibited greater influence than ethnicity on both the composition and diversity of the gut microbiota. In addition, a total of 15 shared biomarkers at the genus level were identified in three distinct locations, including seven in Inner Mongolia, seven in Xinjiang and one in Gansu. Furthermore, 65 unique biomarkers were found in 14 population groups, which indicated specific communities in different populations. Based on the gut microbiota species, two main enterotypes—namely Prevotella (ETP) and Bacteroides (ETB), which consist of *Prevotella* and *Bacteroides* as the core bacterial genus, were observed in Chinese populations. Our unique experimental design using the same ethnic group—Han, as a control in different locations, enables delineating the importance of geographical location and ethnicity on the gut microbiota, and provides the fundamental characteristics of gut microbiota diversity in Chinese populations.

## 1. Introduction

The gut microbiota is considered an internalized “microbial organ” that exerts far-reaching impacts on human health [1]. It helps to extract energy from a wide range of host-indigestible carbohydrates, to produce vitamins, short-chain fatty acids, to promote immune homeostasis and to prevent colonization of the gut by pathogens [2,3,4]. These functions are health-relevant and are considered as ecosystem services that are provided by the gut microbiota for the benefit of the host [5].

Healthy individuals generally maintain a relatively stable structure of the gut microbiota, while microbiome dysbiosis is associated with adverse health conditions that includes inflammatory bowel disease [6], type 2 diabetes mellitus [7], obesity [8], asthma, autism and rheumatoid arthritis [9]. The specific compositions of the microorganisms that constitute the gut microbiota, related to diseases incidence and progression, have been previously considered as bacterial biomarkers [10,11], which might serve as important targets for disease intervention.

Identification of specific microorganisms as key ecosystem service providers and a deeper understanding of the interactions among microorganisms with their hosts, are fundamental issues deserving of further study in human microbiome research [12]. Numerous studies of the gut microbiota composition showed that multiple factors drive substantial differences across individuals in terms of the composition and diversity of the gut microbiota [13,14]. These factors include host genome composition, geographical location, diet and lifestyle [13,15].

There are studies that have focused on whether the existence of stable bacterial biomarkers from the gut microbiota can separate populations from different geographical locations or distinct ethnicities [14,16]. Studies have shown that ethnic background was strongly associated with a variety of microbial taxa, gene families and metabolic pathways; however, this could be explained as being associated with differential dietary intake and other habits common to those sub-populations of individuals from distinct geographical locations [15,17]. It remains unclear which specific factor plays a dominant role in shaping the gut microbiome [18].

China occupies a significant land mass with extensive differences in both climate and environment, and has developed a diverse dietary culture over time [12]. The Han Chinese people and another 55 different ethnic minority groups constitute approximately 20 percent of the world’s population. Traditionally, each ethnic group dwelled in a distinct geographical location and preserved their own dietary habits and specific lifestyles [12,19], especially in the context of Mongolians, Uyghurs and Tibetans. Ethnicity might thus contribute to differences in the diversity of the gut microbiota. However, there are relatively few studies that have analyzed gut microbiota in different geographical locations and ethnicities in China, especially studies that have explored differences between Hans and non-Hans in the same geographic location.

In this study, we investigated the gut microbiota of healthy subjects from seven different geographical locations in China. In each location, we sampled both Han Chinese and one representative ethnic group. With this design, we assessed the impact of geographical location, ethnicity and other factors on the constitution of the gut microbiota, and further quantified predominant factors. We also report on core communities of the Chinese population and bacterial biomarkers of various ethnic groups in different geographic locations. This work might help advance our understanding of gut microbiota diversity in the Chinese population.

## 2. Materials and Methods

### 2.1. Study Groups

We recruited 394 healthy individuals, aged from 17 to 49 years that were unrelated volunteers, consisting of 14 groups from 7 cities in different provinces across China. The subjects were identified as Han and another 7 minority ethnic groups (i.e., the Bai, Hui, Korean, Miao, Mongol, Tibetan and Uyghur, Appendix A). Each non-Han minority ethnic group was recruited from distinct geographical locations with different natural environmental conditions and economic status. The identity and ethnicity of each volunteer was confirmed by evaluating consistency within the census register, spoken language, dressing style and lifestyle with that of the corresponding ethnic group. For comparison, we also recruited Han volunteers from matched locations where each non-Han minority ethnic group was recruited.

None of the volunteers had gastrointestinal tract disorders or had taken any antibiotics for at least 2 months prior to the sampling. The study was carried out in accordance with the recommendations of the local Ethics Review Board on Biomedical Research Involving Human Subjects of the National Health Commission of the People’s Republic of China, with written informed consent from all subjects in accord with the Declaration of Helsinki. The protocol was approved by The Ethics Committee of the China Agricultural University (Research Project Identification No. 2011/23).

The data and sample collection were performed from May to September in 2012. Subjects were required to record their defecation frequency, stool consistency [20], and diet over the past three days. Subjects were instructed to collect fresh fecal samples by themselves with a pre-provided sampling kit (including a foam box, a screw cap tube with RNAlater solution, a tube rack, a pre-cooling ice pack), and immerse the specimen in the screw cap tube with RNAlater solution. Samples were transferred to the local collaboration laboratory within 4 h and stored at 4 °C. After all samples were collected, they were transported on ice to a Beijing laboratory, and stored at 4 °C until analysis.

Geographical locations and environmental data (Appendix A) were obtained from global climate data (WorldClim) and used in bioclimatic modeling (CliMond). WorldClim is recognized by the international community as a regional and accurate climataological database [21]. CliMond provides different formats of environmental data, modeling tools, and meteorological expertise for ecological studies [22].

### 2.2. Analysis of Fecal Microbiota by 16S rRNA Sequencing

We amplified 16S rRNA genes using a broad-range, bacteria-specific primer pair 967F: CAACGCGAAGAACCTTACC; 1046R: CGACAGCCATGCANCACCT for the V6 hypervariable 16S RNA region [23]. Sequencing was carried out by using the Illumina High-seq 2000 platform according to standard procedures.

Operational taxonomic unit (OTU) classification, chimera removal, tree construction and taxonomic assignment were all performed using the QIIME2 package [24], after removing low-quality sequences from the raw datasets according to the provided manual by Illumina. The OTUs mapped against the SILVA 138 SSU/132 LSU database [25], and KEGG Orthologies (KOs) were annotated according to an analysis pipeline of the Tax4Fun R package [26].

### 2.3. Feature Selection

There were 122 factors that might influence the composition of gut microbiota from the questionnaire and WorldClim information (Appendix A). We divided these factors into eight categories that included the following: population indicators, climatic factors, personal physical conditions, cereal foods, foods with high protein levels, vegetables, fruits and drinks. The population indicators included geographical location and ethnicity and the climatic factors were acquired from WorldClim according to the latitude and longitude coordinates. The personal factors reflected the physical conditions of healthy respondents. The other five categories are self-explanatory. In order to explore the interactions between geographical location and ethnicity on dietary habits, we employed quantitative variables that were decomposed by variable decomposition. In order to explore the most important factors that influenced the composition of the gut microbiota, all factors were screened by a redundancy analysis (RDA) and a forward feature selection strategy. Finally, independent factors with the highest R-square were retained.

To further study the effects of factors at taxonomic levels (core OTUs, core genera, OTUs, genera, families, orders, classes and phyla) of the gut microbiota and their Kos, we elected to use permutational multivariate analysis of variance (PERMANOVA) to retrieve the explanatory variance.

### 2.4. The α-Diversity and β-Distance

Shannon Wiener indices were calculated with the aim of evaluating diversity in all individuals based on OTUs. Differences in the Shannon Wiener index among different groups were tested by Turkey-HSD.

In order to verify the clustering of 14 groups at the genus level, we calculated the Bray–Curtis distance for all group-pairs. Similarly, the weighted Unifrac distance and Euclidean distance were used to measure the difference among groups at the OTU and KO levels, respectively. The 196 combinations (14 by 14) were further grouped into four categories: (1) pairs of different ethnic groups and different locations; (2) pairs of same ethnic group but different locations; (3) pairs of different ethnic groups but same location; and (4) pairs of same ethnic group and same location. Differences in distances among the four categories were tested by Turkey-HSD. All analyses were performed with the “vegan” R package [27] and the “GuniFrac” R package [28].

### 2.5. Enterotypes and Bacterial Biomarkers Analysis

Based on the taxonomic composition at the genus level, gut microbiota structures were divided into three enterotypes, which included Prevotella (ETP), Bacteroides (ETB), and Firmicutes (ETF), as described previously [29,30]. The chi-square test was used to explore whether the distribution of enterotypes was influenced by geographical location and ethnicity, i.e., to explore the influence of geographic locations by testing Hans from different geographical locations, and to explore the influence of ethnicity by testing Hans and non-Hans from the same geographical location. 

In order to pick up informative bacterial biomarkers across different groups, the relative abundance of the bacterium was compared among groups by Linear discriminant analysis Effect Size (LEfSe), which generates a composite output of a bacterial biomarker list based on three statistical methods including: the Kruskal–Wallis test, Wilcoxon rank sum test and the linear discriminant analysis [31]. LEfSe was used for comparing the taxonomic characteristics among Han as compared non-Han populations; Hans as compared Hans in different locations; non-Hans as compared non-Hans in different locations; and Hans to non-Hans in the same location. Please refer to the supplemental material for detailed methodological steps and data (Appendix A).

All genera were assessed for their impact on community composition by RDA. We selected the top 50 RDA with the largest impact on community composition. Meanwhile, all genera were analyzed by Spearman’s correlation analysis to derive correlated genera with an absolute value of the Spearman’s rho coefficient > 0.5 and *p*-value < 0.05. Venn diagrams were created to show overlaps for bacterial biomarkers, core genera, the top 50 RDA and the correlated genera.

## 3. Results

### 3.1. Overall Gut Microbiota Distribution in Chinese Populations

At the level of the Operational Taxonomic Unit (OTU), a total of 41,572 OTU types were identified. Han individuals from Guizhou province contain the highest average of OTU types (i.e., 7615 types identified in 360,229 OTU reads, Appendix A), while Han individuals from Xinjiang had the lowest average of OTU types (i.e., 4045 types identified in 101,026 reads, Appendix A). Moreover, 159 core OTUs were detected in all individuals (average of proportion equal to 73.04%, and ranging from 42.18–86.07%, Appendix A, Appendix A). 

At the genus level, 85 core genera were detected in all individuals (mean proportion equal to 85.43%, and ranging from 74.30–93.23%, Appendix A, Appendix A). *Prevotella*, including 19% of *Prevotella*_9, 2.7% of *Prevotella*_2 and 0.22% of *Prevotella*_1 (Appendix A), as well as *Bacteroides* (average proportion equal to 14.37%, Appendix A) were the two most abundant genera found in the gut microbiota.

At the phyla level, Firmicutes, Bacteroidetes, and Proteobacteria constituted the three most dominant bacterial phyla, with 49.08%, 42.06% and 5.39% of the total sequences identified, respectively. The ratio of Firmicutes:Bacteroidetes showed considerable variation among ethnic groups (1.23 ± 0.34, Appendix A), which was consistent with previous studies conducted in Asian, African and Western populations [4,12,17].

At the genetic-function level, a total of 6410 KEGG Orthologies (KOs) were detected when using Tax4Fun (Appendix A). K02014, K06147 and K02004 were three of the most abundant KOs (with an average abundance of 0.0116, 0.00883 and 0.00601, respectively) in all 14 groups (Appendix A). These three KOs were annotated to regulate the iron complex outer membrane receptor proteins, ATP-binding cassette (ABC) subfamily B, and the bacterial putative ABC transport system permease proteins, respectively (Appendix A).

### 3.2. Geographical Location and Ethnicity Were Major Factors Influencing Composition of the Gut Microbiota

We further investigated the factors influencing the composition of gut microbiota. The redundancy analysis (RDA) showed that geographical location and ethnicity influenced the composition of the gut microbiota to a greater extent than all other factors (the adjusted R square was 0.187 and 0.163, respectively, Appendix A). According to variance decomposition, these two factors might indirectly influence the microbiota by affecting dietary habits, especially in cereal foods (Appendix A). In addition, types of fermented dairy products significantly influenced the microbiota with a relatively small adjusted R square of 0.061. This is interesting because the Hans prefer consumption of commercial yogurt, while homemade yogurt is more favored by minorities such as Uygur (Figure 1A, Appendix A).

At each taxonomic level and KOs, it was found that a combination of geographical location and ethnicity explained the incidence of more microbiota variations than could be explained when geography and ethnicity were considered separately (Appendix A, Appendix A). Moreover, geography exerted a greater impact on the top 20 percent of abundant bacteria than did ethnicity, which indicated its important effects on the microbiota (Appendix A, Appendix A).

Although we showed that geographical location and ethnicity both had significant influences on the microbiota at different taxonomic levels and KOs (Appendix A, Appendix A)—when using one-way analysis of covariance (ANCOVA) or one-way permutational multivariate analysis of variance (PERMANOVA)—it was also found that two-way ANCOVA or two-way PERMANOVA analyses only showed a significant influence for the geographic location, but not for ethnicity (Appendix A, Appendix A). 

Taken together, our results indicate that geography and ethnicity might jointly explain the diversity of the gut bacterial composition, while geographical location has a greater influence than ethnicity on the dominant gut bacterial community.

### 3.3. Geographical Location Has a Greater Influence than Ethnicity on α-Diversity and β-Distance

We found that the α-diversity of the gut microbiota, as measured by the Shannon Wiener indices, significantly varied among geographical locations (F = 12.70, *p* < 0.001), but no significance was observed among different ethnic groups (F = 1.13, *p* = 0.34, Figure 2A, and Appendix A), indicating that geographic location more greatly influenced α-diversity of gut microbiota than ethnicity. Hans (5.26 ± 0.18; Figure 2A) and Mongolians (5.26 ± 0.24, Figure 2A) from Inner Mongolia had the highest α-diversity value, while Bai populations from Yunnan showed the lowest value of α-diversity (4.78 ± 0.19, Figure 2A).

As shown by the dendrogram constructed on the basis of the Bray–Curtis distance at the genus level, the 14 ethnic groups were divided into six clusters: (1) Han/Mongolian from Inner Mongolia; (2) Han/Korean from Jilin; (3) Han/Hui from Gansu, and Han from Xinjiang; (4) Han/Miao from Guizhou, and Bai from Yunnan; (5) Han from Tibet/Yunnan; and (6) Tibetan from Tibet and Uygur from Xinjiang (Figure 1B). The first five clusters were most likely affected by geographical factors with adjacent locations.

The influence of geographic location and ethnicity on the composition of gut microbiota and KOs were further assessed by weighted UniFrac distances and the Euclidean distance, respectively. At the OTU level, different ethnic groups from different locations (DL_DE) had the highest weighted UniFrac distances; DL_DE and DL_SE were significantly higher than the same ethnic group from the same geographic location (SL_SE) (Figure 2B, and Appendix A). Similar results were obtained in the comparison of KOs by Euclidean distance. In addition, DL_DE was significantly higher than the other three categories (Figure 2B, and Appendix A). There was a tendency that indicated that geographic location exerted a greater influence than ethnicity did.

### 3.4. Prevotella and Bacteroidetes Contribute to Differences of Enterotype, Community Composition and Diversity of Gut Microbiota

Previous studies showed that Enterotypes of Prevotella (ETP), Bacteroides (ETB), and Firmicutes (ETF) were the three primary enterotypes found in different human populations [29,30]. Our results indicated that the enterotype ETF was rare in Chinese populations, and only one Mongolian among all subjects studied belonged to ETF (Figure 3A). Moreover, both ETP and ETB were two predominant enterotypes, and ETP had a higher frequency than ETB in all groups studied (Figure 3A).

There was no significant difference among Han populations from different geographical locations (χ^2^ = 4.779, *p*-value = 0.572) with respect to enterotypes. Interestingly, the ratio of enterotypes was found to vary among different ethnicities (Figure 3A). Furthermore, within the same geographical location, the percentage of ETB enterotypes in Hans was consistently higher than that found in respective minorities, with the exception of Hui in Gansu province, with some locations reaching statistical significance (e.g., Bai and Tibetan, Yunnan: χ^2^ = 7.179, *p* = 0.007, Tibet: χ^2^ = 4.647, *p* = 0.031, Figure 3A).

As shown in Figure 3B, distributions of *Prevotella*_9 and *Bacteroides* varied between the Han and non-Han minority ethnic groups when assessing the data for overall comparisons. Specifically, the scores for *Prevotella*_9 and Prevotellaceae in the Han groups were significantly lower than those of non-Han groups (Figure 3B, Linear discriminant Analysis: LDA score (log10) > 4), whereas patterns for *Bacteroides* and Bacteroidetes were in the opposite direction (Figure 3B, LDA (log10 score) < −4). In addition, we found that the ratio of *Prevotella*_9:*Bacteroides* was negatively correlated with the Shannon Wiener index (rho = −0.724, *p* < 0.001). Additionally, it was found that Hans and Mongolians from Inner Mongolia had the highest Shannon Wiener indices with a relatively low *Prevotella*_9:*Bacteroides* ratio (Figure 2A). It suggested that the closer the relative abundance of the two dominant genera, then the greater the α-diversity of the gut microbiota will be.

### 3.5. Bacterial Biomarkers of Various Ethnicity in Different Geographical Location

Subjects from the same geographical location or same ethnicity tended to cluster together in the dendrogram based on LDA scores by the LDA effect size analysis (LEfSe) model. This suggested that subjects within the same geographical location or from the same ethnicity might share specific biomarkers (Figure 4A, Appendix A).

All bacterial biomarkers were divided into shared and unique biomarkers according to whether those were detected for the Hans in all, or in particular geographical locations. According to the relative abundance, they were divided into bacterial biomarkers with greater quantity and bacterial biomarkers with decreased quantity. In this way, we evaluated the influence of geographical and ethnic factors on these biomarkers.

In total, 158 bacterial biomarkers including 78 bacterial biomarkers at the genus level were detected by LEfSe (Appendix A). In particular, these included 41 shared biomarkers and 120 unique biomarkers (Appendix A). In terms of the relative abundance, the biomarkers comprised 129 with higher quantity, 14 biomarkers with decreased quantity, and 15 biomarkers with opposite quantitative directions in different locations (Appendix A). In addition, most bacterial biomarkers were found in Xinjiang and Inner Mongolia by LEfSe (Figure 4A and Appendix A, Appendix A), indicating the characteristics of the microbiota in these two provinces.

A total of 15 shared biomarkers at the genus level were detected, including seven (i.e., Coprococcus3, Ruminococcus1, Romboutsia, Parasutterella, Faecalibacterium, Alistipes, Prevotella_9) in Inner Mongolia, seven3 (Pseudomonas, Tissierella, Shewanella, Ignatzschineria, Acholeplasma, Proteiniphilum, Blautia) in Xinjiang, and only one (Prevotella_7) in Gansu (Appendix A). It was found that there was no significant difference in the relative abundance of the majority of shared biomarkers (Appendix A). It suggested that non-Hans individuals showed stronger regional characteristics of shared biomarkers than did Hans in the geographical location. Meanwhile, a total of 65 unique biomarkers for 14 groups at the genus level were detected (Appendix A). We found that the unique biomarkers were mainly derived from Bacteroidetes, Firmicutes and Proteobacteria.

### 3.6. Bacterial Biomarkers Were Correlated with Each Other in Each Communities

Using RDA, it was shown that bacterial biomarkers at the genus level with overlapping top nine RDAs were capable of highly explanatory evidence aligned to the community composition (Figure 4B and Appendix A). For example, *Prevotella*_9, as the most dominant genus, was distributed in all subjects; *Bacteroides* showed higher RDA explanatory power in the microbial community for Inner Mongolia and *Succinivibrio* showed higher RDA explanatory power in Xinjiang (Figure 4B, Appendix A). These results indicated that the bacterial biomarkers not only showed differences among themselves in various groups, but also influence on the whole microbial community structure.

We found that 64 bacterial biomarkers were correlated with each other, including 267 pairs of a positive correlation (Spearman’s rho > 0.5), and 36 pairs that showed a negative correlation relationship (Spearman’s rho < −0.5, Figure 4C, and Appendix A). The negative correlations were found mostly for comparisons between Bacteroidetes and Firmicutes. This suggested that higher abundance bacteria may restrain low abundance bacteria. However, some bacteria with lower abundances like *Actinobacteria*, *Epsilonbacteraeota* and *Spirochaetes*, showed a positive correlation with each other (Figure 4C, and Appendix A), which might suggest an additional mechanism of cooperation in these bacteria.

## 4. Discussion

We collected stool samples of Hans and non-Hans individuals from seven cities of different provinces across China. Based on high throughput sequencing data, we analyzed the composition of gut microbiota and the potential influence of 122 factors. We found that the geographic location and ethnicity most significantly influenced the gut microbiota. Moreover, we further detected core bacteria and bacterial biomarkers among different geographical locations and ethnicities in China. 

We found that differences in the gut microbiota were mainly influenced by geographical location. Many prior studies of different populations had shown that both composition and diversity of the gut microbiota could be influenced by many factors, including underlying diseases, dietary factors, host gene interactions, life styles, geographical location, and climate [4,15,17,32]. Screening out the most critical factor from many factors requires appropriate dimension reduction methods. Geographical location and ethnicity were screened out as the most critical factors by two rounds of RDA and forward selection analysis (Figure 1A, and Appendix A). We further showed that geography explained most of the observed variations in the gut microbiota, as well as other diversity measures. These results are consistent with some prior reports [33]. These studies also verified that geographic factors more strongly influenced the gut microbiota than did genetic, ethnic or other factors among Israeli, Moroccan, Turk, Ghanaian, and Surinamese populations, using the kinship test, PERMANOVA, multidimensional scaling etc. [14,16]. Geographic factors included climatological factors, topography, and regional dietary habits, which need to be further studied.

We consider two potential factors may drive the ethnic differences in gut microbiota composition. First, genotype may contribute to the microbiota changes. Host genotype had important effects on gut microbiota, as shown in previous studies. This study found that Tibetans and Uygurs both had significant difference in gut microbial composition, compared with Hans who were from Tibet or Xinjiang respectively. As shown in the genetic relationship, Tibetan and Uygur had great genetic distance to Han [15,34]. This could contribute to different bacterial colonization in the gut [15,34]. Second, dietary habits may be involved in the ethnic differences. We found that home-made yoghurt and Tibet buttermilk correlated with microbial changes in Uygur and Tibetan populations, respectively. These are traditional fermented foods containing lactic acid bacteria, which could affect the microbial profile. However, ethnicity might be a blend factor, and further studies should identify the potential effective driven elements behind the ethnicity.

The main enterotypes in the Chinese population were ETP and ETB, which was consistent with studies conducted on African populations, but it was different from studies conducted on European or American populations in which ETB and ETF were major enterotypes [15,34]. The ratio of ETP was higher than that of ETB for all ethnic groups or geographical locations. Dietary habits could significantly contribute to enterotypes, since ETP was adept at digesting carbohydrates and fibers [35], while ETF was adept at digesting animal protein and saturated fats in a “Western” diet [36]. Compared with the Western population, high carbohydrate/fiber intake in the Chinese population might formally account for ETP [37].

We identified two key bacterial biomarkers that distinguished Hans and non-Hans populations—including *Prevotella*_9 and *Bacteroides*—which corresponded to the two major enterotypes, ETP and ETB. In addition, *Prevotella* and *Bacteroides* might directly lead to different community compositions and diversity. *Prevotella* tends to change the host environment in terms of digesting carbohydrates and fiber [35], while *Bacteroides* tend to form a community for digesting fats and proteins [38]. According to an extension of the niche theory, if the two dominant genera reach a certain balanced state, it will release increased abundance of the niche allowing the associated bacterium to survive in the host gut micro-environment [5,30,39]. Thus, the overall diversity of the gut microbiota will be increased.

The microbiota composition of the local Hans population was different across different geographical locations but became closer to the non-Hans population in the same location. However, there were no shared biomarkers between the Tibetans and the Hans in Tibet, an observation which was consistent with previous studies in which there was a large difference in the gut microbiota between these two populations [40,41]. It may be due to the highland environment that prevents dietary communication between Tibetans and Hans. Tibetans prefer highland barley as cereal food, while the local Hans do not necessarily (significant intaking frequency between Tibetans and Hans, *t* = 12.932, *p* < 0.001 by two sample *t*-test methods, Appendix A). Meanwhile, genetic differences between Tibetans and Hans could also contribute to the microbial differences [40,41]. In general, however, with increased residence time, biomarkers with regional or ethnic characteristics are becoming similar, due to the mutual acceptance of a dietary culture.

In conclusion, through the comparative analysis between Hans and non-Hans populations, this study found that geography had the most important influence on gut microbiota. Moreover, there was a shared influence among different factors. Therefore, further studies are warranted to explore the internal and external factors of variations in the gut microbiota, and to establish network interactions of various potential factors. We found Prevotella_9 and Bacteroides to be the two dominant genera of gut bacteria in Chinese people. These two genera, and other associated bacterium as bacterial biomarkers, built a correlated microbiotic system, and then formed the two main enterotypes, ETP and ETB, which collectively provide a theoretical reference for regional intestinal diagnoses and treatment. In addition, by using this approach, more bacterial biomarkers could be found, with their applications gradually evolving from classification to targeted medical uses.

## Figures and Tables

**Figure 1 microorganisms-08-01579-f001:**
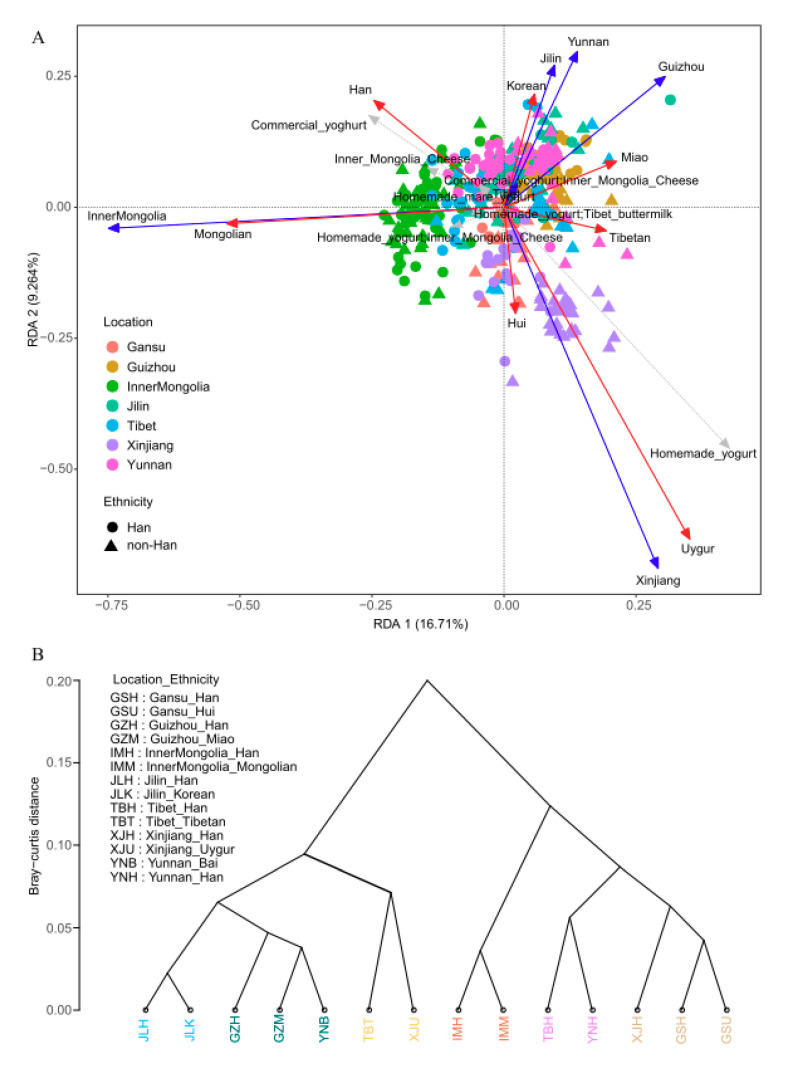
(**A**) Redundancy analysis (RDA) biplot (RDA on a covariance matrix, scaling 2) showing the two first axes of a canonical ordination of 394 samples. Circles and triangles represent different ethnicity [Han and other minorities (non_Han)] with 7 colors used to identify geographic location; only three factors (geographic location by using blue arrows, ethnicity by using red arrows and types_of_fermented_dairy_products by using grey arrows) were retained from 122 potential influencing variables after the forward selection. (**B**) Complete linkage agglomerative clustering dendrogram of Bray–Curtis distance of genera among different groups. The *X*-axis labels in different colors represent different groups.

**Figure 2 microorganisms-08-01579-f002:**
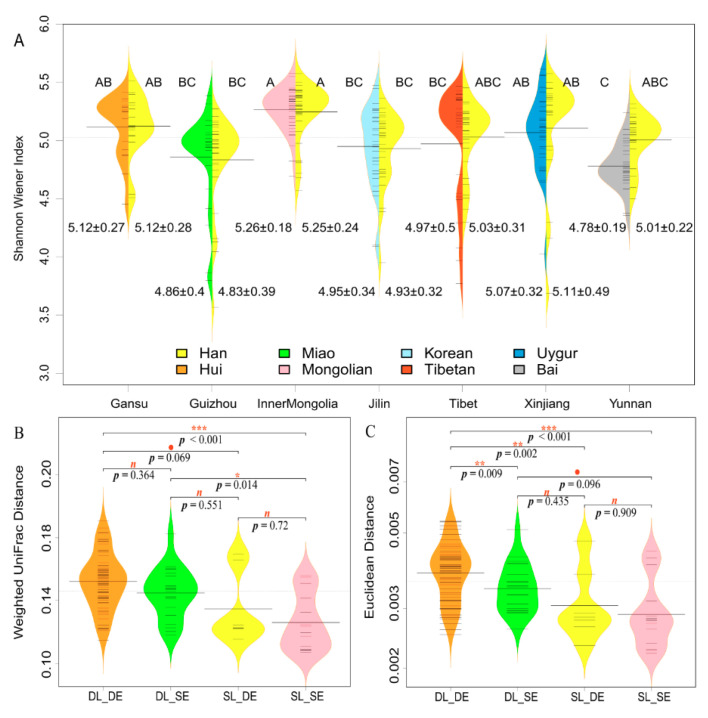
(**A**) Beanplot of the Shannon Wiener index. Different letters mean significant difference at the 5% level, and the same letter means no significant difference. From “A” to “C”, the Shannon Wiener index decreased significantly. Tail of each plot notes mean value ± standard deviation. (**B**) and (**C**) show, respectively, the difference of weighted UniFrac distance of operational taxonomic units (OTUs) and the Euclidean distance of KOs among different combinations with geographic location and ethnicity (at the *X*-axis, DL = Different Location; SL = Same Location; DE = Different Ethnicity; SE = Same Ethnicity); “*p*” represents *p*-value of Tukey tests between two groups connected by horizontal lines. ***: *p* < 0.001; **: 0.001 ≤ *p* < 0.01; *: 0.01 ≤ *p* < 0.05; ●: 0.05 ≤ *p* < 0.1; n: *p* ≥ 0.1, not significant.

**Figure 3 microorganisms-08-01579-f003:**
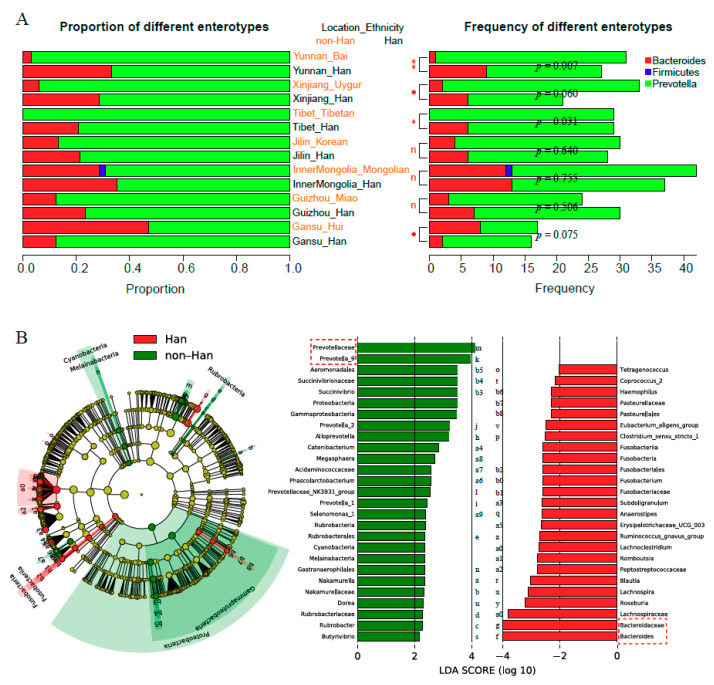
(**A**) Gut microbiota structures were divided into three enterotypes, Prevotella (ETP); Bacteroides (ETB); Firmicutes (ETF), based on their taxonomic composition at genus level; “*p*” represents the *p*-value of chi-square tests between two groups connected by longitudinal lines. **: 0.001 ≤ *p* < 0.01; *: 0.01 ≤ *p* < 0.05; ●: 0.05 ≤ *p* < 0.1; n: *p* ≥ 0.1, no significant. (**B**) Plot_Cladogram of LEfSe analysis. Taxonomic representation of statistically and biologically consistent differences between Han groups and non-Han groups. Differences are represented in the color of the most abundant class (red indicating Han groups, green indicating non-Han groups and yellow non-significant). Each circle’s diameter is proportional to the taxon’s abundance. The genera, which is significantly different between Hans and non-Hans, and its taxonomic rank are both highlighted—for example, multiple differentially abundant sibling taxa consistent with the variation of the parent clade; the right list of LEfSe provides features that were differential among conditions of interest with statistical and biological significance, ranking them according to the effect size.

**Figure 4 microorganisms-08-01579-f004:**
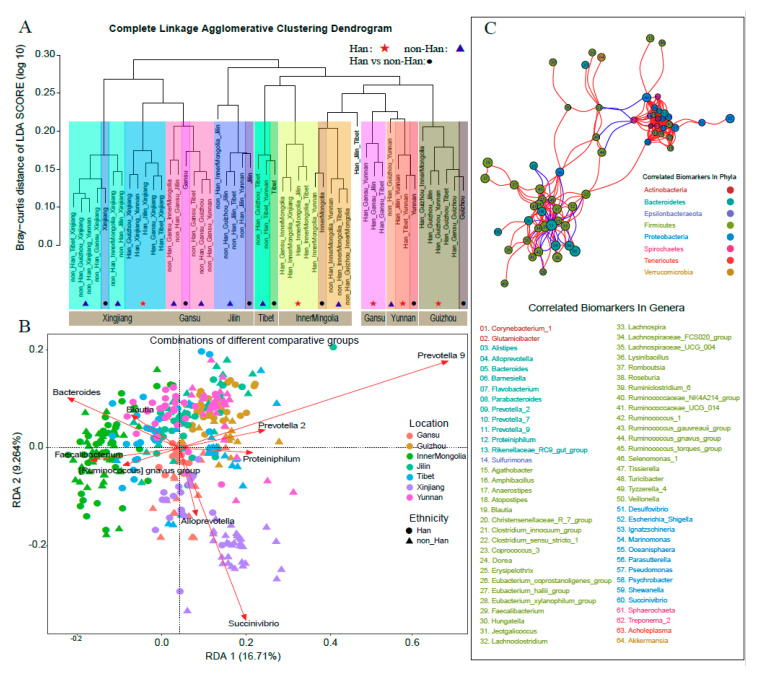
(**A**) Complete linkage agglomerative clustering dendrogram of the Bray–Curtis distance of the LDA score (log 10) between combinations of different comparative groups. The *X*-axis in brown marks the major geographic location effects. Different geographic or ethnic groups are represented in the other shading colors. ★, ▲ and ● represented corresponding to the comparison results of different categories. (**B**) RDA biplot (RDA on a covariance matrix, scaling 2) showing the two first axes of a canonical ordination of 394 samples. Circles and triangles represent different ethnicities [Han and other minorities (non_Han)] with 7 colors to identify geographic location; 655 genera (just showed Top 9 significant genera by using red arrows). (**C**) Co-occurrence network deduced from all genera (Appendix A) enriched in 64 bacterial biomarkers which were correlated each other. Sizes of the nodes represent the logarithm of the total number of each genus sequence. Red edges, Spearman’s rank correlation coefficient > 0.5, adjusted *p* < 0.05; blue edges, Spearman’s rank correlation coefficient < −0.5, adjusted *p* < 0.05. Different colors of nodes represent the corresponding phyla. Each node with a number annotated to the genera name were given the corresponding annotation.

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
