# Peer review of "The Core and Distinction of the Gut Microbiota in Chinese Populations across Geography and Ethnicity"

_microorganisms, 2020, doi:10.3390/microorganisms8101579_

Round 1

Reviewer 1 Report

The authors well characterized the microbiota of different Chinese population and found that geographic location and ethicity were the main determinants of diversity.

A few factors should be further analyzed to substain their conclusion:

  1. BMI and rate of overweight and obesity is well known to influence the intestinal microbiota.  Please provide the data and related microbiota in the different Chinese population
  2. The intake of yogurt or other fermented foods or probiotics need to be considered
  3. Life style, levels of physical activity and socio-economic factors should be' better described and analyzed 

Author Response

The authors well characterized the microbiota of different Chinese population and found that geographic location and ethicity were the main determinants of diversity.

A few factors should be further analyzed to substain their conclusion:

  1. BMI and rate of overweight and obesity is well known to influence the intestinal microbiota. Please provide the data and related microbiota in the different Chinese population

Thank you for your comments. We agreed with this reviewer that BMI had significantly impact on gut microbiota, as shown in previous studies. In this study, actually, we did considered BMI as one factor to investigate its influencing the composition of gut microbiota (BMI raw data showed in Table S1A,B). The results showed that BMI had less influence in gut microbiota compared with geographical location and ethnicity (adjusted R square was 0.002291 in Table S2A). This indicated that geographical location and ethnicity were primary factors you that the BMI variation in different subject groups may affect gut microbiota. Meanwhile, the BMI variation as one of Personal physical condition was observed in different subject groups. So we using forward selection methods to assess the effects of BMI and other factors of Personal physical condition on microbiota in all group. The results showed that Weight, as a highly correlated factor with BMI and obesity to some extent, could impact microbial communities rather than BMI itself (Table S2B). Meanwhile, we also found Geographic Location and Ethnicity had more influencing in microbiota (adjusted R square was 0.19989 and 0.1777 respectively). The results could be found in supplementary Table S2A.

  1. The intake of yogurt or other fermented foods or probiotics need to be considered.

Thank you for your comments. In this study, we found some volunteers took fermented foods (pickles) or yoghurt based on dietary records over past three days (Table S1B). We did considered these potential probiotics-contained food as one of factors to investigate their effects on microbiota. We found that types of fermented dairy products significantly influenced the microbiota with a relatively small adjusted R square of 0.0607. The results were showed in Figure 1A and Table S2A.

  1. Life style, levels of physical activity and socio-economic factors should be' better described and analyzed.

Thank you for your comments. We agreed with this reviewer that life style could affect microbiota. Life style was considered as a complex blend factor including physical activity, diet, economic condition, cultural habit et al. These factors were usually correlative, which were difficult to evaluate their contribution to life style independently. In this study, we recruited healthy individuals from 7 distinct geographical regions inhabited by ethnic minorities. Subjects from different regions had distinctive life style. We record 122 factors (including population indicators and diet factors) that might influence the composition of gut microbiota from the questionnaire and WorldClim information (Table S1B). The interactions of these factors and their impacts to microbiota were analyzed (Table S2A). Although this study did not collect physical activity or economic data of each subjects, geographical location, ethnicity and the climatic factors were considered as population indicators which could represent the life style of different populations.

Reviewer 2 Report

The manuscript entitled: “The core and distinction of the gut microbiota in Chinese populations across geography and ethnicity”, by Lin et al studies the gut microbiota in al large group of 394 healthy Chinese individuals, consisting of 14 groups from 7 cities in different provinces across China exploring the effect of ethnicity versus geography on gut microbiota.

The paper shows similarities with the paper from Zhang et al: A phylo-functional core of gut microbiota in healthy young Chinese cohorts across lifestyles, geography and ethnicities. ISME J. 2015 Sep; 9(9): 1979–1990. Though none of the authors/Institutes are the same. Who set-up the study and collected the samples, in which year were samples collected? Please clearly state the novelty of the current paper in comparison to the Zhang paper. If this is an extended evaluation, it should be clearly stated and original researchers should be acknowledged.

Please find my remarks below:

-The study must have been a logistic challenge, authors described that “samples were transferred to the laboratory within 4 h, shipped on ice and stored at 4ºC until analysis”. Is this to a local laboratory or to the laboratory in Beijing? Please described samples collection and transport more in detail to garantuee the quality of the samples collected over such a large geographically area. Please specify when the samples were collected (year?).

-Although this is a healthy population, do authors have in on medication, drinking and smoking habits, educational level of the volunteers?

-It is not clear to the reviewer what the potential drivers of gut microbiota composition within an ethnicity are, based on the current data.

-Authors do provide raw data on diet, but it is not clear to the reviewer how this was incorporated into the current analyses, and what the conclusions are (e.g. for fibre intake).

-Paragraph 3.5 and 3.6, please add that these are bacterial biomarkers

-Discussion: p11/14 line 380-384: The explanation for the lack of shared biomarkers between Tibetans and Hans in Tibet is not very convicing: “It may be due to the highland environment that prevents dietary communication between Tibetans and Hans. For example, Tibetans prefer highland barley as cereal food, while the local Hans do not necessarily.” Do the authors have data to support this?

Author Response

The manuscript entitled: “The core and distinction of the gut microbiota in Chinese populations across geography and ethnicity”, by Lin et al studies the gut microbiota in al large group of 394 healthy Chinese individuals, consisting of 14 groups from 7 cities in different provinces across China exploring the effect of ethnicity versus geography on gut microbiota.

  1. The paper shows similarities with the paper from Zhang et al: A phylo-functional core of gut microbiota in healthy young Chinese cohorts across lifestyles, geography and ethnicities. ISME J. 2015 Sep; 9(9): 1979–1990. Though none of the authors/Institutes are the same. Who set-up the study and collected the samples, in which year were samples collected?

Thank you for your comments. This study is an independent study which is not related to the paper of Zhang’s et al. (2015). We preformed subject recruitment and sample collection from May to September of 2012, and DNA was extracted from June to November of 2012.

  1. Please clearly state the novelty of the current paper in comparison to the Zhang paper. If this is an extended evaluation, it should be clearly stated and original researchers should be acknowledged.

Thank you for your comments. Zhang et al. (2015) recruited similar size of healthy young adults from 7 ethnic groups living in 9 provinces in China. In each province, they match rural and urban subjects to investigate the microbial differences of populations living in rural and urban. They recruited minorities in regions inhabited by ethnic minorities, including Mongol in Inner Inner Mongolia, Kazakh in Xinjiang, Uyghur in Xinjiang, Tibetan in Tibet, Zhuang in Guangxi, Bai in Yunnan.

In our study, we recruited Han and other 7 minorities from 7 cities in different provinces across China. We match Han volunteers from matched locations where each non-Han minority ethnic group were recruited. By this design, we could identify geographical effects on microbiota by assessing Han’s microbiota in different regions. We also could evaluate the ethnic effects by comparing Han and non-Han minority in each location. Our unique design using Han, as a control in different locations, enables delineating the importance of geographical location and ethnicity on the gut microbiota, and provides the fundamental characteristics of gut microbiota diversity in Chinese populations. We made the statements about the novelty in Introduction part (P2, Line 67-70, Line 72-76).

  1. The study must have been a logistic challenge, authors described that “samples were transferred to the laboratory within 4 h, shipped on ice and stored at 4ºC until analysis”. Is this to a local laboratory or to the laboratory in Beijing? Please described samples collection and transport more in detail to garantuee the quality of the samples collected over such a large geographically area. Please specify when the samples were collected (year?).

Thank you for your comments. We apologized for less detailed description about the sampling. We performed data and sample collection from May to September in 2012. In this study, we have collaboration laboratory in each location which are in local universities or CDC. Subjects were instructed to collect fresh fecal samples by themselves with a pre-provided sampling kit. Then samples were transferred to the local collaboration laboratory by subject themselves or team members within 4 hr. After all samples were collected in each region, they were transported to Beijing lab and stored until analysis. We have revised the manuscript for more details (P3, Line 95-102).

  1. Although this is a healthy population, do authors have in on medication, drinking and smoking habits, educational level of the volunteers?

Thank you for your comments. We excluded volunteers had gastrointestinal tract disorders or had taken any antibiotics for at least 2 months prior to the sampling based on questionnaire. We have considered avoiding the impact of medication history, and volunteers with chronic diseases and recent medication were excluded from the beginning of investigation. We also recorded drinking of each volunteer, and considered them as factors in the analysis (Table S1B). However, we did not get smoking data of subjects. A recent study showed that smoking or smoking cessation leads to minor changes in the intestinal microbiota (Sublette et al., 2020). Previous study showed that educational level mainly impacted the diet choices (Lê et al., 2013), which further affected the microbiota. Although, we did not get the educational level data, we analyzed the dietary effects on microbiota.

  1. It is not clear to the reviewer what the potential drivers of gut microbiota composition within an ethnicity are, based on the current data.

Thank you for raising this interesting topic. We think two factors may drive the ethnic differences. First, genotype may contribute to the microbiota changes. Host genotype had important effects on gut microbiota, as shown in previous studies. This study found that Tibetan from Tibet and Uygur from Xinjiang had significant variation in microbiota compared with Han in corresponding regions. As shown in genetic relationship, Tibetan and Uygur had great genetic distance to Han (Yao et al., 2002). This could contribute to different bacterial colonization in gut (Org et al., 2015). Second, diet habit may be involved in ethnic differences. We found that home-made yoghurt and Tibet buttermilk correlated with microbial changes of Uygur and Tibetan, respectively. These were traditional fermented food containing lactic acid bacteria, which could affect the microbial profile. However, ethnicity might be a blend factor, and further studies should identify the potential effective driven elements behind the ethnicity. We added discussion in the manuscript (P11, Line 365-375).

  1. Authors do provide raw data on diet, but it is not clear to the reviewer how this was incorporated into the current analyses, and what the conclusions are (e.g. for fibre intake).

Thank you for your comments. Based on the dietary information in questionnaire, we clustered the diet into 5 categories including cereal foods, foods with high protein levels, vegetables, fruits and drinks. We assigned the variables based on types of food in each category and calculated the intaking frequency of each subjects. Then we used these datasets as variables to assess their effects on microbiota. Detailed information was in Table S1.

  1. Paragraph 3.5 and 3.6, please add that these are bacterial biomarkers

Thank you for your suggestion. We revised the biomarkers as ‘Bacterial biomarkers’ in the manuscript.

  1. Discussion: p11/14 line 380-384: The explanation for the lack of shared biomarkers between Tibetans and Hans in Tibet is not very convicing: “It may be due to the highland environment that prevents dietary communication between Tibetans and Hans. For example, Tibetans prefer highland barley as cereal food, while the local Hans do not necessarily.” Do the authors have data to support this?

Thank you for your comments. We calculated the intaking frequency of highland barley in Tibetans and Hans in Tibet. Tibetans ate much higher amount of highland barley than Hans (Tibetans vs Hans, t = 12.932, p < 0.001 by Two Sample t-test methods). On the other hand, host genetic differences between Tibetans and Hans may also contribute to the microbial differences. We revised the manuscript (P11, Line 398-401).

References

Sublette, M.G.; Cross, T.-W.L.; Korcarz, C.E.; Hansen, K.M.; Murga-Garrido, S.M.; Hazen, S.L.; Wang, Z.; Oguss, M.K.; Rey, F.E.; Stein, J.H. Effects of Smoking and Smoking Cessation on the Intestinal Microbiota. J. Clin. Med. 2020, 9, 2963.

Le, J.; Dallongeville, J.; Wagner, A.; Arveiler, D.; Haas, B.; Cottel, D.; Simon, C.; Dauchet, L. Attitudes toward healthy eating: a mediator of the educational level–diet relationship. European Journal of Clinical Nutrition. 2013, 7. 808-814

Yao, Y. G.; Nie, L.; Harpending, H.; Fu, Y. X.; Yuan, Z. G.; Zhang, Y.P. Genetic relationship of chinese ethnic populations revealed by mtdna sequence diversity. American Journal of Physical Anthropology.2002, 118(1), 63-76

Org, E.; Parks, B. W.; Joo, J. W.; Emert, B.; Schwartzman, W.; Kang, E. Y.;Mehrabian, M.; Pan, C.; Knight, R.; Gunsalus, R.;Drake, T. A.; Eskin, E.; Lusis, A. J. Genetic and environmental control of host-gut microbiota interactions. Genome research.2015, 25(10), 1558–1569.

Round 2

Reviewer 1 Report

I appreciate the efforts the authors made to answer my queries. No other substantial issue to raise

Reviewer 2 Report

I reviewed the revised version of the manuscript entitled: “The core and distinction of the gut microbiota in Chinese populations across geography and ethnicity”, by Lin et al. The authors addressed my remarks in a satisfactory way.